# Phase angle is independently associated with muscle strength across multiple handgrip strength metrics in young adults: A cross-sectional study

Juan Carlos Calderón-González[1], Luis Hebert Palma-Pulido[1], Gonzalo Romero-Martínez[1], Juan Carlos Urriago-Fontal[1], María Elisa Álvarez-Ossa[1], Frank Carrera-Gil[2], Robinson Ramírez-Vélez[1,3,4]*

1 Facultad de Ciencias de la Educación, Unidad Central del Valle del Cauca (UCEVA), Tuluá, Colombia, 2 Department of Food and Nutrition, Faculty of Health Sciences, Pontificia Universidad Javeriana Seccional Cali, Cali, Colombia, 3 Navarrabiomed, Hospital Universitario de Navarra (HUN), Universidad Pública de Navarra (UPNA), Instituto de Investigación Sanitaria de Navarra (IdiSNA), Pamplona, España, 4 CIBER of Frailty and Healthy Aging (CIBERFES), Instituto de Salud Carlos III, Madrid, Spain

* robin640@hotmail.com, robinson.ramirez@unavarra.es

## Abstract

### Objective

Phase angle derived from bioelectrical impedance analysis has been proposed as a marker of muscle quality associated with muscle function; however, the extent to which its association with muscle strength is influenced by health status, dietary patterns, physical activity, and demographic factors remains incompletely characterized. This study examined the association between phase angle and muscle strength assessed using multiple handgrip strength (HGS) metrics, accounting for relevant clinical and lifestyle factors.

### Methods

This cross-sectional study included 1,125 adults with complete data on phase angle, skeletal muscle mass, and HGS. Phase angle was measured using bioelectrical impedance analysis. Muscle strength was assessed as absolute HGS and HGS normalized to height² (HGS/h²). Low muscle strength was defined using sex- and age-specific international normative values below the 10th percentile for both HGS and HGS/height².

### Results

Phase angle was consistently associated with muscle strength across both continuous and dichotomous analyses. In multivariable linear regression models, phase angle was positively associated with HGS ($\beta_{std}$ = 0.26; P < 0.001) or HGS/ h² ($\beta_{std}$ = 0.38; P < 0.001). In logistic regression models, higher phase angle was associated with lower odds of low muscle strength (in fully adjusted Model 2 HGS: OR,

**Data availability statement:** The data underlying this study cannot be made publicly available because they contain sensitive personal and clinical information that could compromise participant confidentiality. Given the sample size and the specificity of the recruitment setting, there is a risk of participant re-identification. In addition, the manuscript includes contextual details (e.g., institution, location, and recruitment period) that may increase this risk when combined with the dataset. Data sharing is therefore restricted in accordance with the study protocol and the informed consent approved by the institutional Research Ethics Board. De-identified data may be made available upon reasonable request to the Research Ethics Board at Unidad Central del Valle del Cauca (UCEVA), Tuluá, Colombia (vicerrectori-ainv@uceva.edu.co), for researchers who meet the criteria for access to confidential data.

**Funding:** This work was supported by the Unidad Central del Valle del Cauca, Tuluá, Colombia ("UCEVA IMPACTA 2020-2030" CONVOCATORIA INTERNA No. 18 VIGENCIA AÑO 2025). The funding organisations had no role in the design and conduct of the study; in the collection, analysis, and interpretation of the data; or in the preparation, review, or approval of the manuscript.

**Competing interests:** The authors have declared that no competing interests exist.

0.37; 95% CI, 0.29–0.48; HGS/h²: OR, 0.31; 95% CI, 0.24–0.41; both P<0.001). These associations were independent of sex, age, body fat percentage, physical activity level, and comorbidity categories.

## Conclusions and relevance

Phase angle was consistently associated with muscle strength across continuous and dichotomous outcomes after adjustment for adiposity, physical activity, and comorbidities. These findings suggest that phase angle may have potential utility as a supportive population-level marker in similar young adult populations; however, it should not be used as a substitute for direct muscle strength assessment at the individual level.

## 1. Introduction

Muscle strength has received increasing attention as a health indicator, given its associations with functional capacity and life expectancy [1]. It is a defining phenotypic feature of sarcopenia [2] and sarcopenic obesity [3], and has been consistently linked to wasting conditions [4], metabolic diseases [5], disability, and both all-cause and cardiovascular mortality [6]. Accurate measurement of muscle strength is therefore relevant for both risk stratification and clinical diagnosis. Grip dynamometry is the most widely used proxy measure due to its simplicity, low cost, portability, and prognostic value [7]. Its use is limited, however, in patients with upper-limb impairment, reduced cooperation, or impaired consciousness. Other approaches—manual muscle testing, handheld dynamometry, and functional performance tests—are constrained by examiner dependence or the need for adequate mobility, which motivates interest in alternative accessible indicators.

Bioelectrical impedance analysis (BIA) is a noninvasive, portable method that requires minimal participant cooperation and can be applied in both clinical and field settings. Phase angle, derived from BIA as the arctangent of reactance to resistance at a given frequency, reflects cellular membrane properties and hydration status, with higher values generally interpreted as indicating greater cellular integrity and mass [8]. Phase angle has been associated with measures of muscle mass and strength across a range of populations, and values below population-specific thresholds have been proposed as indicators of reduced muscle quality and increased risk of sarcopenia [9–11].

Despite a growing body of evidence linking phase angle to body composition and physical performance, several limitations in the existing literature remain. First, large-scale studies (n>500) evaluating the association between phase angle and HGS in healthy young adults are lacking, despite this age range representing the period of peak muscle development and a key window for preventive interventions [9]. Second, most available data are derived from European and East Asian populations, and the extent to which these findings generalize to Latin American populations remains uncertain [10]. Third, prior studies have not examined the phase angle–HGS association while simultaneously accounting for multiple lifestyle and cardiometabolic factors at the individual level [11,12].

The present study addresses these gaps in a large Colombian university cohort, using multi-frequency segmental BIA, standardized dynamometry, and validated assessments of lifestyle and health-related factors. We hypothesized that phase angle would be positively associated with HGS across both absolute and normalized metrics, independent of body composition and lifestyle factors.

## 2. Methods

### 2.1. Study design and participants

This study, part of the EpiHealth-UCEVA project, examined the relationships among lifestyle habits, physical fitness, and mental health outcomes in university students using a cross-sectional design. The study adhered to the Strengthening the Reporting of Observational Studies in Epidemiology (STROBE) guidelines. In total, 1,200 undergraduate students were invited to participate. The participants were recruited from a single public university using convenience sampling through institutional emails, the university website, social media announcements, and on-campus posters and flyers. The inclusion criteria were an age of 18–35 years (corresponding to a period of higher risk for incident mental health problems) and enrollment in an undergraduate academic program during the 2024 or 2025 academic year. Students from all programs and semesters were eligible. After excluding 75 students because of incomplete data, the final analytical sample comprised 1,125 participants. The study protocol was approved by the Ethics Committee on Human Research (protocol no. ID 3103−2025) from Unidad Central del Valle del Cauca (UCEVA) in Tuluá, Colombia. The studies were conducted in accordance with the local legislation and institutional requirements. The participants provided their written informed consent to participate in this study. The recruitment period lasted from 06 June 2025–09 December 2025.

### 2.2. Physical assessments

Standing height was measured to the nearest 0.1 cm using a calibrated stadiometer (Seca, Chino, CA, USA). Body mass index (BMI) was calculated as weight divided by height squared. Waist circumference was measured at the midpoint between the lowest rib and the iliac crest. Body composition was assessed using multifrequency bioelectrical impedance analysis (InBody 770, Biospace, Seoul, Korea). The device provided estimates of fat-free mass, body fat percentage, and related indices using proprietary algorithms. Phase angle was calculated as $\arctan[Xc/R] \times 180/\pi$. All measurements were performed by a single trained technician under standardized conditions: morning sessions (07:00–10:00 h), minimum 2-hour fast, no vigorous activity in the preceding 12 hours, bladder voiding immediately prior, bare feet and palms on electrode surfaces, light clothing without metal accessories, arms abducted approximately 15° during the ~45-second measurement. The InBody 770 uses direct segmental multi-frequency BIA (DSM-MFBIA) at six frequencies (1, 5, 50, 250, 500, and 1,000 kHz), which reduces susceptibility to hydration variability compared to single-frequency devices [13,14]. Published test-retest reliability for this device is excellent (ICC 0.94–0.99 for phase angle and body composition variables [13–15]), and intra-rater variability was limited by the fixed-contact electrode design and single-technician protocol. Internal measurement quality was tracked using the InBody Score (mean $71.0 \pm 7.0$; 64% of participants scored ≥70). Intra-study repeated measurements were not obtained, which is acknowledged as a limitation. Blood pressure was measured twice using a digital device (OMRON, Kyoto, Japan), and the average was recorded. Assessments were performed with participants seated after at least 5 minutes of rest, using an appropriately sized cuff placed on the upper arm. Systolic and diastolic blood pressures (mmHg) were recorded, and mean arterial pressure was calculated as one-third of the pulse pressure plus the diastolic blood pressure.

HGS was measured with an analog dynamometer (Takei 5001; Takei Scientific Instruments). Participants stood with arms fully extended and without trunk contact. Each participant completed three trials per hand, alternating hands, and the best value per hand was recorded. The higher of the two hands was taken as absolute HGS. HGS normalized to height² (HGS/h²) was calculated as a size-independent index [16]. Low muscle strength was defined using two reference frameworks: (1) sex-specific cutoffs derived from a Colombian normative dataset [17], and (2) sex- and age-specific values

below the 10th percentile based on international normative data for young adults (18–39 years) [18]. Colombian reference thresholds were examined in sensitivity analyses. These approaches were applied to evaluate the robustness of classification across different normative standards.

## 2.3. Measures of lifestyle behaviors, demographics, and covariables

All questionnaires were administered under standardized conditions following established protocols. Physical activity was assessed using the short form of the International Physical Activity Questionnaire (IPAQ), a self-administered instrument that captures the frequency and duration of vigorous, moderate, and walking activities performed during the previous 7 days [19]. Data were processed according to the standardized IPAQ scoring protocol, with each activity category converted to metabolic equivalent task minutes per week (MET-min/week) using established coefficients (vigorous, 8.0; moderate, 4.0; walking, 3.3). Total physical activity was calculated as the sum across activity domains and expressed as MET-min/week. Participants were classified as having low, moderate, or high physical activity according to established thresholds [19]. Diet quality was assessed using the validated 14-item Mediterranean Diet Adherence Screener (MEDAS), developed for the PREDIMED study [20]. The instrument uses binary (yes/no) responses to assess adherence to key dietary components, yielding a total score ranging from 0 to 14, with higher scores indicating greater adherence. Consistent with prior studies, a score of 8 or higher was used to define adequate adherence to the Mediterranean diet [21]. Sociodemographic characteristics were collected by means of a standardized self-report questionnaire and included marital status (common-law union, single, married, or separated), socioeconomic status classified according to national criteria (low [levels I–II], medium [level III], or high [levels V–VI]), place of residence (urban or rural), race or ethnic group (Mestizo, Afro-descendant, or Indigenous/other), smoking status, and alcohol consumption (yes or no). Self-reported medical conditions were categorized as respiratory, cardiovascular, metabolic, mental, or other. Participants could report more than one condition; those who reported no physician-diagnosed conditions were classified as having no disease.

## 2.4. Statistical analysis

The sample size was primarily determined by cohort availability and ensured adequate precision for multivariable modeling. The primary outcome was the Pearson correlation coefficient (r) between PhA and HGS. The expected effect size was estimated conservatively at r = 0.42, based on correlations reported in comparable cross-sectional studies: Simón-Frapolli et al. [22] reported r = 0.42 (n = 75); Ballarin et al. [23] reported r = 0.45 (n = 229); Rodríguez-Rodríguez et al. [12] reported r = 0.58 (n = 223); and Cioffi et al. [24] reported r = 0.54 (n = 140). Using a two-tailed test with $\alpha = 0.05$ and minimum power of 80% ($1 - \beta \geq 0.80$), the required minimum sample size was n = 43. The enrolled sample of n = 1,125 substantially exceeded this threshold, for detecting the expected effect size (r = 0.42, $\alpha = 0.05$, bilateral test), and sufficient power to detect even small correlations of $r \geq 0.09$ (87.7% power). This large sample was intentional, as secondary objectives included subgroup analyses stratified by sex, physical activity level, and health status — each requiring adequate within-stratum power. Calculations were performed using the pwr.r.test function (pwr package, R). Descriptive statistics were used to summarize participant characteristics by sex. Prior to analysis, normality was assessed using the D'Agostino–Pearson $K^2$ omnibus test. Therefore, Spearman's rank correlations were used as primary association measures, with confidence intervals computed via Fisher's Z-transformation. Continuous variables were expressed as means and standard deviations or medians and interquartile ranges, as appropriate, and categorical variables as counts and percentages. Differences by sex were assessed using Mann–Whitney U tests for continuous variables. For categorical variables, comparisons were performed using the $\chi^2$ test or Fisher's exact test. All tests were two-sided, and a P value of <0.05 was considered statistically significant. No imputation of missing data was performed; analyses were based on available cases. To examine associations between phase angle, body composition, and muscle strength, sex-stratified correlation analyses were conducted. Spearman's rank correlations ($\rho$) were calculated to assess relationships between phase angle, body composition variables, and muscle strength measures (HGS and HGS/height$^2$). Multivariable linear regression models were used to examine the associations of phase

angle with continuous measures of muscle strength (HGS and HGS/height$^2$). Logistic regression models were used to assess associations with low muscle strength, defined using sex- and age-specific cutoffs [17,18]. Because phase angle was strongly correlated with BIA-derived body composition measures (skeletal muscle mass: $\rho=0.738$; skeletal muscle index: $\rho=0.808$; fat-free mass: $\rho=0.720$), and these variables are derived from the same underlying resistance and reactance measurements, they were not included simultaneously in regression models. Primary models used body mass index, calculated from anthropometric measurements independent of BIA, as the adiposity covariate. Models substituting BMI with skeletal muscle index or percentage body fat are presented as sensitivity analyses. This approach was used to reduce structural collinearity (variance inflation factor <2.7 in all primary models) while acknowledging that associations between phase angle and BIA-derived body composition measures may partly reflect shared computational inputs. Model assumptions were evaluated using standard residual diagnostics. Although phase angle cutoffs of 4.2° to 4.5° have been proposed to identify low muscle quality, these thresholds were derived primarily from older or clinical populations [25]. Given the young adult composition of the present sample (mean [SD] phase angle, 5.79 [0.78]°; <8% with values <4.5°), phase angle was modeled as a continuous variable to preserve statistical power and to minimize potential misclassification associated with unvalidated thresholds. Sensitivity analyses using tertile-based categories yielded consistent results (Supplementary Figure S4 in S1 File). Sample size calculations were performed in R (pwr package), whereas all primary statistical analyses were conducted in SPSS Statistics version 26.0 (IBM Corp).

## 3. Results

### 3.1. Study population

The study included 1,125 participants (638 women, 487 men) (Table 1). Sociodemographic characteristics were similar between sexes with respect to marital status, socioeconomic status, residence, and academic field. Men were older (20.7±3.9 vs. 19.9±2.9 years), taller (1.73±0.07 vs. 1.60±0.06 m), and heavier (71.4±14.4 vs. 61.3±12.1 kg). They also had greater muscle mass (31.2±5.1 vs. 21.3±3.4 kg), higher phase angle (6.38±0.69° vs. 5.34±0.57°), higher absolute HGS (37.7±8.7 vs. 24.2±5.2 kg), slightly higher MEDAS scores (6.0 ± 1.9 vs. 5.6 ± 2.0), whereas the prevalence of high Mediterranean diet adherence was similar between sexes. Women had higher body fat percentage (34.6±6.6 vs. 21.4±7.2%). Physical activity levels were predominantly low or moderate in both groups.

### 3.2. Correlations between phase angle, body composition, and muscle strength

The correlations observed in the full sample were largely attributable to between-sex differences in muscle mass and phase angle (Table 2), rather than within-sex associations. The highest correlations were observed for fat-free mass index ($\rho=0.781$; $P<0.001$), skeletal muscle mass ($\rho=0.738$; $P<0.001$), and fat-free mass ($\rho=0.720$; $P<0.001$). These associations likely reflect, at least in part, shared methodological features inherent to BIA rather than independent biological relationships. Within sex strata, correlations between phase angle and absolute HGS were moderate in women ($\rho=0.332$; $P<0.001$) and men ($\rho=0.350$; $P<0.001$). Correlations with HGS/height² were modestly stronger (women: $\rho=0.383$; men: $\rho=0.435$; both $P<0.001$). Fisher z tests indicated no evidence of a difference between sexes for correlations with absolute HGS (Z= −0.338; $P=0.736$) or normalized handgrip strength (Z= −1.043; $P=0.297$). Sensitivity analyses confirmed that the phase angle–HGS association was robust and consistent across all self-reported disease subgroups ($\rho$ range: 0.499–0.682, all $P<0.01$) and all physical activity strata ($\rho$ range: 0.585–0.648, all $P<0.001$), indicating that the relationship is not confined to healthy or physically active populations, Supplementary Table S2-S3 in S1 File.

### 3.3. Association of phase angle with muscle strength parameters

Multiple linear regression models used all continuous predictors standardized to z-scores (mean=0, SD=1) to allow direct comparison of effect magnitudes (Table 3). In Model 1 (adjusted for sex, age, and BMI), phase angle was positively

**Table 1. Characteristics of the study population stratified by sex.**

| Characteristic | Women (n = 638) | Men (n = 487) | P Value |
|---|---|---|---|
| **Marital status, n (%)** | | | |
| Common-law union | 35 (5.5) | 30 (6.2) | 0.568 |
| Single | 591 (92.6) | 440 (90.3) | |
| Married | 9 (1.4) | 14 (2.9) | |
| Separated/ Divorced | 3 (0.5) | 3 (0.6) | |
| **Socioeconomic status, n (%)** | | | |
| Low (Levels I–II) | 434 (68.0) | 327 (67.1) | 0.286 |
| Medium (Level III) | 156 (24.5) | 130 (26.7) | |
| High (Levels V–VI) | 48 (7.5) | 30 (6.2) | |
| **Residence, n (%)** | | | |
| Urban | 531 (83.2) | 416 (85.4) | 0.600 |
| Rural | 107 (16.8) | 71 (14.6) | |
| **Race/ethnicity, n (%)** | | | |
| Mestizo | 561 (87.9) | 413 (84.8) | 0.073 |
| Afro-descendant | 8 (1.3) | 15 (3.1) | |
| Indigenous/Other | 69 (10.8) | 59 (12.1) | |
| **Self-report diseases, n (%)** | | | |
| Respiratory | 38 (6.0) | 33 (6.8) | 0.003 |
| Cardiovascular | 13 (2.0) | 10 (2.1) | |
| Metabolic | 60 (9.4) | 30 (6.2) | |
| Mental | 59 (9.2) | 21 (4.3) | |
| Other conditions | 24 (3.8) | 12 (2.5) | |
| None | 444 (69.6) | 380 (78.2) | |
| **Anthropometry/ body composition** | | | |
| Age, mean (SD), y | 19.96 (2.96) | 20.75 (3.99) | <0.001 |
| Weight, mean (SD), kg | 61.32 (12.10) | 71.47 (14.46) | <0.001 |
| Height, mean (SD), m | 1.60 (0.06) | 1.73 (0.07) | <0.001 |
| Body mass index, mean (SD), kg/m$^2$ | 24.04 (4.32) | 23.94 (4.38) | 0.710 |
| Skeletal muscle mass, mean (SD), kg | 21.38 (3.40) | 31.23 (5.13) | <0.001 |
| Fat-free mass, mean (SD), kg | 39.51 (5.65) | 55.53 (8.54) | <0.001 |
| Body fat, mean (SD), % | 34.68 (6.62) | 21.40 (7.21) | <0.001 |
| Fat-free mass index, mean (SD), kg/m$^2$ | 15.48 (1.72) | 18.58 (2.24) | <0.001 |
| Fat mass index, mean (SD), kg/m$^2$ | 8.56 (3.06) | 5.36 (2.77) | <0.001 |
| Phase angle, mean (SD), degrees | 5.34 (0.57) | 6.38 (0.60) | <0.001 |
| **Blood pressure** | | | |
| Systolic blood pressure, mean (SD), mm Hg | 113.55 (12.99) | 121.52 (15.03) | <0.001 |
| Diastolic blood pressure, mean (SD), mm Hg | 70.81 (10.17) | 73.11 (10.40) | <0.001 |
| Mean arterial pressure, mean (SD), mm Hg | 85.06 (9.71) | 89.25 (10.32) | <0.001 |
| **Muscular strength** | | | |
| Absolute HGS (mean), mean (SD), kg | 24.27 (5.24) | 37.73 (8.73) | <0.001 |
| Normalized HGS (HGS/height$^2$), mean (SD) | 9.53 (2.00) | 12.67 (2.88) | <0.001 |
| **Diet** | | | |
| MEDAS score | 5.6 (2.0) | 6.0 (1.9) | <0.001 |
| ≥ 8 High adherence, n (%) | 55 (8.6) | 44 (9.0) | 0.832 |
| **Alcohol intake, n (%)** | 9 (1.4) | 17 (3.5) | 0.027 |

*(Continued)*

**Table 1.** (Continued)

| Characteristic | Women<br>(n = 638) | Men<br>(n = 487) | *P* Value |
|---|---|---|---|
| **Physical activity levels, n (%)** | | | |
| Low | 327 (51.3) | 126 (25.9) | <0.001 |
| Moderate | 135 (21.2) | 127 (26.1) | |
| High | 173 (27.1) | 229 (47.0) | |
| Missing data | 3 (0.3) | 5 (1.0) | |
| MET-min/wk score | 1759.7 (1493.9) | 2427.5 (1613.9) | <0.001 |

**Legend:** Values are presented as mean (SD) for continuous variables and numbers (percentages) for categorical variables; BP, blood pressure; ME-DAS: 14-item Mediterranean Diet Adherence Screener; Total physical activity was expressed as metabolic equivalent–minutes per week.

**Table 2. Spearman correlation profile of Phase Angle by sex and full sample.**

| Variable | Overall ρ<br>(n = 1,125) | Female<br>ρ (n = 638) | Male<br>ρ (n = 487) | p-diff |
|---|---|---|---|---|
| **Muscle strength** | | | | |
| Absolute HGS (kg) | 0.657*** | 0.332*** | 0.350*** | 0.736 |
| Normalized HGS (HGS/h²) | 0.611*** | 0.383*** | 0.435*** | 0.297 |
| **Body composition and anthropometrics** | | | | |
| Fat-free mass index (kg/m²) | 0.781*** | 0.618*** | 0.618*** | 0.998 |
| Skeletal muscle mass (kg) | 0.738*** | 0.454*** | 0.410*** | 0.374 |
| Fat-free mass (kg) | 0.720*** | 0.415*** | 0.381*** | 0.500 |
| Body fat (%) | −0.487*** | −0.004 | −0.021 | 0.767 |
| Body fat mass (kg) | −0.195*** | 0.157*** | 0.101* | 0.343 |
| Age (y) | 0.241*** | 0.192*** | 0.301*** | 0.055 |
| Body mass index (kg/m²) | 0.280*** | 0.388*** | 0.429*** | 0.412 |
| **Lifestyle** | | | | |
| Total MET·min/week | 0.350*** | 0.252*** | 0.202*** | 0.378 |
| MVPA (min/week) † | −0.181*** | −0.051 | 0.047 | 0.106 |
| Sedentary behavior (min/day) | 0.031 | −0.014 | 0.006 | 0.740 |
| Mediterranean diet (MEDAS) score | 0.095** | 0.063 | 0.016 | 0.438 |
| Alcohol intake | 0.069* | 0.021 | 0.053 | 0.590 |
| **Blood pressure** | | | | |
| Systolic blood pressure (mm Hg) | 0.314*** | 0.144*** | 0.232*** | 0.128 |
| Diastolic blood pressure (mm Hg) | 0.169*** | 0.156*** | 0.075 | 0.173 |

**Legend:** Correlations of phase angle with fat-free mass index, skeletal muscle mass, fat-free mass, and body fat are expected to be inflated due to shared BIA measurement basis. †The inverse MVPA correlation should be interpreted cautiously and may reflect differences in variable derivation, distribution, or residual reporting error; Total MET captures all activity types whereas MVPA only captures moderate to vigorous minutes. No sex × variable interaction was statistically significant (all p-diff > 0.05), indicating that PhA–variable associations did not differ significantly between females and males after Fisher Z correction. ρ = Spearman rank correlation coefficient. Significance: *** p < 0.001, ** p < 0.01, * p < 0.05. All p-values two-tailed. p-diff: Fisher Z-transformation test comparing ρ between sexes. †Available for n = 1,117 (8 missing IPAQ values).

associated with absolute HGS (β = 2.56 kg per SD; 95% CI: 2.01–3.10; βstd = 0.265; P < 0.001; R² = 0.555) and normalized HGS (β = 1.10 kg/m² per SD; 95% CI: 0.91–1.28; βstd = 0.382; P < 0.001; R² = 0.425). Phase angle was the strongest independent predictor of normalized HGS among the variables in Model 1.

**Table 3. Multivariable Linear Regression Models for Handgrip Strength Metrics.**

| | Absolute HGS (kg) | | | | Normalized HGS (HGS/h²) | | | |
|---|---|---|---|---|---|---|---|---|
| | β | β std | 95% CI | P | β | β std | 95% CI | P |
| **Model 1 — Main effects** | R²=0.555 Adj.R²=0.554 | | | | R²=0.425 Adj.R²=0.423 | | | |
| Phase angle (z) | 2.56 | 0.265 | 2.01–3.10 | <0.001 | 1.10 | 0.382 | 0.91–1.28 | <0.001 |
| Sex (male) | 9.83 | | 0.5058.78–10.88 | <0.001 | 1.59 | | 0.2731.23–1.94 | <0.001 |
| Age (z) | 0.95 | 0.099 | 0.56–1.35 | <0.001 | 0.36 | 0.124 | 0.22–0.49 | <0.001 |
| BMI (z) | 0.73 | 0.076 | 0.32–1.14 | <0.001 | 0.19 | 0.066 | 0.05–0.33 | 0.007 |
| **Model 2 — Interactions** | R²=0.561 ΔR²=0.006 | | | | R²=0.432 ΔR²=0.007 | | | |
| Phase angle (z) | 1.74 | 0.180 | 1.02–2.47 | <0.001 | 0.82 | 0.284 | 0.57–1.06 | <0.001 |
| Phase angle × Sex | 1.82 | 0.188 | 0.78–2.85 | <0.001 | 0.62 | 0.217 | 0.27–0.97 | <0.001 |
| Phase angle × BMI | 0.17 | 0.018 | −0.22–0.56 | 0.391 | 0.05 | 0.016 | −0.09− 0.18 | 0.497 |

**Legend:** β, unstandardized regression coefficient; β std, standardized regression coefficient (continuous predictors entered as z-scores); 95% CI, 95% confidence interval; BMI, body mass index; HGS, handgrip strength; All VIF<2.5, indicating no multicollinearity concern. Phase angle rows are highlighted in bold. **Model 1** includes phase angle, sex, age, and BMI as main effects. **Model 2** additionally includes two interaction terms (phase angle × sex; phase angle × BMI). Sex is coded as a binary indicator (reference: female). Continuous predictors (phase angle, age, BMI) were standardized to z-scores (mean=0, SD=1) prior to entry; unstandardized β values therefore represent the change in HGS per 1-SD increment in the predictor. All *P* values are two-tailed.

Adding two interaction terms (phase angle × sex; phase angle × BMI) in Model 2 improved explained variance only marginally (ΔR²=0.006 for absolute HGS; ΔR²=0.007 for normalized HGS). Phase angle remained a significant predictor across all models (all P<0.001). The phase angle × sex interaction was statistically significant (β=1.82; 95% CI: 0.78–2.85; P<0.001) and normalized HGS (β = 0.62; 95% CI, 0.27–0.97; P<0.001), although the increase in explained variance was small). Collinearity was not a concern in any model (all VIF<2.5).

### 3.4. Association of phase angle with low muscle strength

After adjustment for sex, age, and body fat, higher phase angle was associated with lower odds of low absolute HGS (OR 0.35; 95% CI 0.27–0.46; P<0.001; Nagelkerke R²=0.143). This association held after further adjustment for physical activity and comorbidities (OR 0.37; 95% CI 0.29–0.48; P<0.001; Model 2). Findings were consistent for normalized HGS: Model 1 (OR 0.29; 95% CI 0.22–0.38; P<0.001; Nagelkerke R²=0.168) and Model 2 (OR 0.31; 95% CI 0.24–0.41; P<0.001). Hosmer–Lemeshow tests did not indicate poor model calibration (all P>0.10). No evidence of substantial multicollinearity was identified in any model (Table 4).

## 4. Discussion

In this cross-sectional study of university students, phase angle derived from multi-frequency bioelectrical impedance analysis was consistently associated with muscle strength across multiple analytical approaches. Correlation analyses showed that these associations were attenuated after stratification by sex, indicating that full-sample estimates were largely influenced by between-sex differences rather than within-sex associations. In multivariable linear models, phase angle remained associated with both absolute and normalized HGS after adjustment for measured covariates. Although a statistically significant interaction with sex was observed, its contribution to explained variance was small and unlikely to be clinically meaningful. Similar patterns were observed in logistic models, in which higher phase angle was associated with lower odds of low muscle strength after adjustment. Overall, these findings indicate that phase angle is consistently associated with muscle strength across analytical strategies and outcome definitions.

**Table 4. Multivariable Associations Between Phase Angle and Low Muscle Strength Metrics.**

| Variable | Model 1 | | | Model 2 | | |
|---|---|---|---|---|---|---|
| **Low HGS[a]** | **OR** | **95% CI** | ***p*** | **OR** | **95% CI** | ***p*** |
| Phase angle (°) | 0.35 | 0.27–0.46 | <0.001 | 0.37 | 0.29–0.48 | <0.001 |
| Sex (male) | 5.78 | 3.61–9.26 | <0.001 | 5.71 | 3.56–9.18 | <0.001 |
| Age (years) | 0.89 | 0.84–0.94 | <0.001 | 0.89 | 0.84–0.94 | <0.001 |
| Body fat (%) | 1.02 | 1.00–1.04 | 0.114 | 1.01 | 0.99–1.03 | 0.165 |
| Physical activity (active) | — | — | — | 0.82 | 0.62–1.10 | 0.194 |
| Self-report diseases (yes) | — | — | — | 1.18 | 0.87–1.60 | 0.291 |
| **Low HGS/h²[b]** | | | | | | |
| Phase angle (°) | 0.30 | 0.23–0.39 | <0.001 | 0.31 | 0.24–0.41 | <0.001 |
| Sex (male) | 4.42 | 2.80–6.98 | <0.001 | 4.46 | 2.81–7.09 | <0.001 |
| Age (years) | 0.91 | 0.87–0.96 | <0.001 | 0.91 | 0.86–0.96 | <0.001 |
| Body fat (%) | 1.01 | 0.99–1.03 | 0.195 | 1.01 | 0.99–1.03 | 0.314 |
| Physical activity (active) | — | — | — | 0.79 | 0.60–1.05 | 0.105 |
| Self-report diseases (yes) | — | — | — | 1.37 | 1.01–1.84 | 0.040 |

[a]Model 1: Adjusted for sex, age, and body fat percentage, n = 1,125 | Events = 335 | Nagelkerke $R^2$ = 0.143, Model 2: Additionally adjusted for physical activity (IPAQ) and self-report diseases, n = 1,116 | Events = 332 | Nagelkerke $R^2$ = 0.144. [b]Model 1: Adjusted for sex, age, and body fat percentage, n = 1,125 | Events = 376 | Nagelkerke $R^2$ = 0.156, Model 2: Additionally adjusted for physical activity (IPAQ) and self-report diseases, n = 1,116 | Events = 372 | Nagelkerke $R^2$ = 0.164. Missing data: Model 2 excludes 9 participants with missing IPAQ (n = 8) or Disease (n = 1) data.

HGS reflects the integrated function of neural, muscular, and connective tissue systems involved in force generation [26]. Phase angle, derived from bioelectrical impedance analysis, is considered a composite indicator of cellular properties related to membrane function and hydration status. It may therefore be associated with aspects of muscle quality relevant to strength performance. Although experimental data have linked phase angle to muscle tissue characteristics, it remains an indirect measure, and the associations observed in this study are best interpreted as reflecting general cellular properties rather than specific biological mechanisms.

Recent diagnostic accuracy data are consistent with a role for phase angle as a marker of muscle impairment. Pooled AUC values of 0.71–0.84 have been reported for sarcopenia detection in older adults [25,27], with optimal cut-offs of 4.2°–4.5°. The lower discriminatory performance found here (AUC 0.66–0.67) was expected: healthy young adults have narrower variation in both phase angle and muscle strength than geriatric or clinical samples. The sex-specific thresholds identified in this study (females ≤5.30°; males ≤6.30°) were notably higher than published clinical cut-offs, which is consistent with the need for age-specific reference values, (see Supplementary Figure S4 and Table S1 in S1 File). These data provide a reference for longitudinal studies aimed at identifying early trajectories of muscle quality decline in this age group.

The large correlations between phase angle and BIA-derived body composition variables (ρ = 0.72 with FFM, ρ = 0.79 with SMI) reflect, in part, a shared measurement basis: all BIA-derived parameters are computed from the same raw resistance (R) and reactance (Xc) values, and phase angle (arctangent of Xc/R) is mathematically related to the indices used to estimate FFM and SMI. Within-sex analyses show that these correlations are driven largely by sex differences in muscle mass, not by independent biological variation. Accordingly, phase angle should not be entered alongside other BIA-derived variables in the same regression model, and sex-stratified reporting is warranted.

These findings support phase angle as a noninvasive marker that may assist in assessing muscle function when direct strength measurements are unavailable at the population level [28–30]. The moderate sex-stratified correlations (ρ = 0.33–0.44), however, indicate that phase angle accounts for roughly 11–19% of strength variance within each sex and cannot substitute for direct HGS assessment in individual clinical evaluation. Its utility is more plausible at the population

level—for screening and health monitoring, particularly in resource-limited settings where dynamometry is not routinely available [29].

Phase angle was consistently associated with muscle strength across different operational definitions of the outcome, although the magnitude of association varied. The larger standardized coefficient for normalized versus absolute HGS (βstd = 0.38 vs. 0.26) suggests that body-size normalization improves the sensitivity of phase angle as a strength indicator. Sensitivity analyses confirmed stability of these associations across disease subgroups (Supplementary Table S2 in S1 File) and physical activity strata (Supplementary Table S3 in S1 File). From a clinical perspective, our findings support phase angle as a noninvasive marker that may assist in the assessment of muscle function when direct strength measurements are unavailable or difficult to perform. This may be particularly relevant in individuals with limited cooperation, impaired consciousness, or physical disability. In clinical contexts such as disease-related malnutrition, phase angle may serve as a functional proxy of neuromuscular integrity, complementing circulating biomarkers such as the C-terminal agrin fragment, which has been associated with low muscle mass and impaired physical function in malnourished [31] and sarcopenic individuals [32]. Within this framework, the consistent association between phase angle and HGS observed in our study suggests that phase angle may capture downstream functional alterations reflected by such biomarkers, without the need for invasive blood sampling.

Our findings indicate that phase angle is independently associated with muscle strength regardless of how strength is expressed. The larger standardized coefficient observed for normalized HGS suggests that height normalization isolates the phase angle signal more cleanly by removing the body-size variance that sex and BMI jointly explain in absolute HGS models. The consistent significance of the phase angle × sex interaction across both outcomes further supports sex-stratified reporting when evaluating this biomarker in young adult populations. Moreover, emerging evidence indicates that both phase angle and HGS are sensitive to short-term nutritional and physical interventions, allowing early identification of patients who respond more effectively to nutritional therapy compared with traditional anthropometric measures such as body weight [33]. In summary, phase angle and HGS appear to capture complementary dimensions of muscle function, reflecting electrical–cellular integrity and mechanical performance, respectively. These characteristics suggest that phase angle may warrant further study as a noninvasive marker for identifying nutrition- and muscle-related phenotypes and for monitoring responses to therapeutic interventions [34].

### 4.1. Limitation

Several limitations should be considered. First, the cross-sectional design precludes causal inference and does not allow assessment of temporal changes in phase angle or muscle strength. Second, all measurements were obtained at a single time point in university students from one institution, which may limit external validity. Third, the use of convenience sampling introduces the possibility of selection bias, and the uneven distribution across academic disciplines may have influenced the sample composition (Supplementary Figure S5 in S1 File). Fourth, although measurements were performed under standardized conditions and the multi-frequency design of the InBody 770 reduces sensitivity to hydration status, residual within-individual variability in hydration cannot be excluded. The extracellular water–to–total body water ratio, an established indicator of hydration status in BIA, was not directly available. Although the consistency of the phase angle–HGS association in the euhydrated subsample (n = 700; ρ = 0.613) suggests limited influence of hydration, residual confounding cannot be excluded. Fifth, the sample was predominantly young (modal age, 19–21 years) and drawn from an urban–semi-urban Colombian university setting, which may limit generalizability to other populations, including rural communities, individuals engaged in manual labor, or those with limited access to health care. Sixth, physical activity was assessed by self-report using the IPAQ-SF, which is subject to recall and social desirability bias and has only moderate validity relative to accelerometry (r = 0.33–0.52 [35]). This limitation may have attenuated observed associations and reduced precision in analyses stratified by physical activity. Finally, intra-session test–retest reliability of bioelectrical impedance measurements was not assessed. Although published reliability estimates for the InBody 770 are high

(intraclass correlation coefficients, 0.94–0.99 [13–15]), population-specific estimates were not available. A small proportion of measurements (<2%) yielded InBody scores below 60, which may indicate suboptimal measurement quality. Taken together, these limitations should be considered when interpreting the magnitude and generalizability of the observed associations.

## 5. Conclusion

Phase angle was consistently associated with muscle strength across multiple analytical specifications. These results support the use of multifrequency BIA-derived phase angle as a potential population-level marker of musculoskeletal health, particularly in settings where direct dynamometry is not available. Whether changes in phase angle over time track clinically meaningful changes in muscle strength remains to be established in longitudinal and interventional studies.

## Supporting information

**S1 File. Supplementary Figure S1. Phase angle vs HGS by sex (overall $\rho = 0.657$, P<0.001.** The overall correlation is shown for descriptive purposes and is partly driven by between-sex differences). Supplementary Figure S2. Phase angle vs physical activity levels (overall trend P<0.001). Supplementary Figure S3. Sensitivity analysis for phase angle by diseases status. Supplementary Figure S4. Exploratory ROC analysis of phase angle for low HGS defined by sex-specific tertiles. Supplementary Figure S5. Sampling distribution by academic field, sex composition, phase angle variation, and phase angle–HGS association (n = 1,125). Supplementary Table S1. Low muscle strength was defined using sex- and age-specific international normative values below the 10th percentile for HGS. Supplementary Table S2. Sensitivity analysis for phase angle–HGS association across self-reported disease status. Supplementary Table S3. Sensitivity analysis for phase angle–HGS association and physical activity levels (IPAQ).
(PDF)

## Author contributions

**Conceptualization:** Juan Carlos Calderón-González, Juan Carlos Urriago-Fontal, Frank Carrera-Gil, Robinson Ramírez-Vélez.

**Data curation:** Juan Carlos Calderón-González, Frank Carrera-Gil, Robinson Ramírez-Vélez.

**Formal analysis:** Robinson Ramírez-Vélez.

**Funding acquisition:** Juan Carlos Calderón-González, Gonzalo Romero-Martínez, Juan Carlos Urriago-Fontal, Robinson Ramírez-Vélez.

**Investigation:** Juan Carlos Calderón-González, María Elisa Álvarez-Ossa, Frank Carrera-Gil, Robinson Ramírez-Vélez.

**Methodology:** Juan Carlos Calderón-González, Luis Hebert Palma-Pulido, María Elisa Álvarez-Ossa, Frank Carrera-Gil, Robinson Ramírez-Vélez.

**Project administration:** Juan Carlos Urriago-Fontal, María Elisa Álvarez-Ossa, Robinson Ramírez-Vélez.

**Resources:** Juan Carlos Urriago-Fontal.

**Supervision:** Gonzalo Romero-Martínez, Robinson Ramírez-Vélez.

**Validation:** Luis Hebert Palma-Pulido, Gonzalo Romero-Martínez, Robinson Ramírez-Vélez.

**Visualization:** Robinson Ramírez-Vélez.

**Writing – original draft:** Juan Carlos Calderón-González, Luis Hebert Palma-Pulido, Gonzalo Romero-Martínez, Juan Carlos Urriago-Fontal, María Elisa Álvarez-Ossa, Frank Carrera-Gil, Robinson Ramírez-Vélez.

**Writing – review & editing:** Frank Carrera-Gil, Robinson Ramírez-Vélez.

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
