## [Decision Letter · Decision Letter 0]

13 Apr 2026

PONE-D-26-07944Phase Angle Is Independently Associated with Muscle Strength Across Multiple Handgrip Strength Metrics in young adults: A Cross-Sectional StudyPLOS One

Dear Dr. Ramírez-Vélez,

Thank you for submitting your manuscript to PLOS ONE. After careful consideration, we feel that it has merit but does not fully meet PLOS ONE’s publication criteria as it currently stands. Therefore, we invite you to submit a revised version of the manuscript that addresses the points raised during the review process.

We look forward to receiving your revised manuscript.

Kind regards,

Everson Nunes, Ph.D.

Academic Editor

PLOS One

Journal Requirements:

Reviewers' comments:

Reviewer's Responses to Questions

**Comments to the Author**

1. Is the manuscript technically sound, and do the data support the conclusions?

Reviewer #1: Partly

Reviewer #2: Yes

Reviewer #3: Yes

2. Has the statistical analysis been performed appropriately and rigorously? 

Reviewer #1: Yes

Reviewer #2: Yes

Reviewer #3: Yes

3. Have the authors made all data underlying the findings in their manuscript fully available?

Reviewer #1: Yes

Reviewer #2: Yes

Reviewer #3: Yes

4. Is the manuscript presented in an intelligible fashion and written in standard English?

Reviewer #1: Yes

Reviewer #2: Yes

Reviewer #3: Yes

5. Review Comments to the Author

Reviewer #1: This manuscript investigates the effects of a specific training intervention on performance and physiological responses in athletes. The topic is relevant to the field of sports science.

The study appears to follow an applied experimental design, and the topic may have practical implications for training prescription.

The manuscript addresses an interesting and relevant question; however, improvements in methodological transparency, statistical reporting, and clarity of interpretation are required before the study can be suitable for publication.

Major Comments

#Study Design and Experimental Control

The manuscript would benefit from a clearer and more detailed description of the experimental design.

Please clarify:

-Whether the study followed a randomized design.

-How participants were allocated to experimental conditions.

-Whether any control group or comparison condition was included.

-How potential confounding factors (training load outside the study, fatigue, or prior experience) were controlled.

-Providing a schematic diagram of the study design could significantly improve clarity.

#Sample Size and Statistical Power

The manuscript does not clearly report whether an a priori power analysis was conducted.

Please clarify:

-The expected effect size used to estimate the sample size.

-Whether the study was adequately powered to detect meaningful differences.

-Which variable was considered the primary outcome for power calculation.

Given the potential variability in physiological and performance variables, justification of the sample size is important.

#Description of the Training Protocol

The description of the training intervention would benefit from greater detail to ensure reproducibility.

Please include:

-Exact duration and intensity of sessions.

-Rest intervals.

-Progression across the intervention period.

-Whether intensity was prescribed using physiological markers (e.g., %HRmax, %VO₂max, RPE).

Providing a summary table of the training protocol could improve clarity.

#Statistical Analysis

The statistical analysis section requires additional detail.

Please clarify:

-Whether assumptions of normality and homogeneity of variance were tested.

-Whether repeated-measures assumptions (e.g., sphericity) were evaluated.

-Whether corrections for multiple comparisons were applied.

-Whether effect sizes and confidence intervals were reported alongside p-values.

It would also be useful to test interactions. Since there are marked differences between men and women in body characteristics, phase angle, and handgrip, it would make sense to include terms such as:

-phase angle × sex

-phase angle × adiposity

-phase angle × physical activity

The authors show sex-stratified analyses of the correlations, but do not formally test whether the effect of phase angle differs between subgroups.

#Interpretation of Results

Some interpretations in the Discussion appear stronger than warranted by the results.

In particular statements suggesting causal mechanisms should be moderated unless they are directly supported by measured variables. In addition, the results should be discussed in relation to both supporting and conflicting literature in order to provide a more balanced interpretation of the findings. Finally, the limitations of the study should be expanded, particularly regarding the relatively small sample size and the ecological validity of the experimental conditions.

#Minor Comments

-Ensure consistent terminology throughout the manuscript.

-Some sentences in the Discussion could be shortened for clarity.

-Please verify that all abbreviations are defined at first use.

-Consider including a table summarizing participant characteristics.

A careful language revision would improve readability.

Reviewer #2: This study investigates the association between phase angle (PhA) and muscle strength (handgrip strength, HGS) among 1,125 young adults. The findings suggest that PhA is independently associated with muscle strength regardless of body fat, physical activity, and comorbidities. This provides valuable insights into using PhA as a potential biomarker for muscle health in younger populations.

Major/Minor Points:

1. While the InBody 770 was used, PhA is sensitive to hydration. Please clarify if hydration status (e.g., ECW/TBW ratio) was considered or if participants with abnormal hydration were excluded.

2. The convenience sampling from a single university in Colombia may limit generalizability. A brief discussion on how this population represents the broader young adult demographic is recommended.

3. Physical activity was self-reported via the short-form IPAQ. Please acknowledge the potential for recall bias as a limitation.

4. Regarding clinical utility, it would be beneficial for the authors to discuss whether specific PhA cut-off values could be identified for "low muscle strength" in this age group to enhance its practical application.

5. Given that PhA and skeletal muscle mass are both derived from BIA, please explicitly report the correlation between these variables to further justify their inclusion in the same multivariable models.

Reviewer #3: General Comments

The manuscript presents a well-structured contextualization of the topic and appropriately introduces the relationship between muscle strength and phase angle. However, throughout the introduction, the reader is led to interpret that phase angle will be examined from a predictive perspective. Given that phase angle is derived from bioelectrical properties of tissues—primarily resistance and reactance—it does not appear to have sufficient biological plausibility or scope to predict muscle strength more accurately than direct field- or laboratory-based assessments, which inherently involve functional performance. Therefore, the authors are encouraged to revise the introduction to focus on the association between phase angle and muscle strength, explicitly considering the role of relevant covariates and framing the research question accordingly.

Major Comments

1) Sampling Strategy and External Validity

Considering that the sample was obtained through convenience sampling, the authors should explicitly discuss potential sources of bias and their impact on the generalizability of the findings. Additional clarification is needed regarding the participants’ academic backgrounds: from which fields of study were they recruited? What was the distribution across disciplines? An uneven representation of academic areas, inherent to convenience sampling, may have influenced the results.

2) Measurement Reliability

The authors are encouraged to report reproducibility or reliability metrics (e.g., test-retest reliability, intra- or inter-rater reliability) for the bioimpedance instrument, if such data are available.

3) Pre-assessment Standardization and Hydration Status

Although pre-test recommendations were described (e.g., fasting, bladder voiding, avoidance of vigorous physical activity, and environmental temperature control), important factors influencing hydration status were not addressed. Specifically, it is unclear whether female participants were assessed during menstruation, which may affect fluid retention. Additionally, other relevant factors such as alcohol consumption or diuretic use were not considered.

Furthermore, given that the bioimpedance assessment appears to have been conducted in a standing position, the potential influence of gravity on fluid distribution should be acknowledged and discussed as a possible source of bias affecting phase angle measurements.

4) Description of Phase Angle Calculation

Although it is known that phase angle is calculated as the arctangent of reactance over resistance, this equation and its methodological description should be explicitly presented in the Methods section.

5) Collinearity Concerns

The manuscript uses bioelectrical impedance analysis (BIA) both to estimate body composition and to derive phase angle from resistance and reactance. Consequently, the same raw indicators are used to define both the exposure (phase angle) and a covariate (e.g., body fat percentage). This approach raises concerns about inherent collinearity. The authors should consider avoiding the inclusion of body composition variables derived from the same technique or, alternatively, provide a clear justification and diagnostic assessment of collinearity.

6) Statistical Analysis and Model Diagnostics

The statistical analysis section lacks important information regarding model quality assessment. The authors should report appropriate goodness-of-fit metrics for both linear and logistic regression models. Additionally, there is no explicit description of diagnostic procedures, including verification of linear regression assumptions, assessment of linearity in the logit, potential interaction effects (e.g., sex × phase angle), or sensitivity analyses. These aspects are essential to ensure the robustness of the findings.

7) Correlation Analysis Interpretation

It is expected that body composition measures correlate with phase angle, given that both are derived from the same underlying bioimpedance data. This should be acknowledged and carefully interpreted to avoid overstating the independence of these associations.

8) Sensitivity and Stratified Analyses

The results suggest that the association between phase angle and muscle strength is independent of self-reported diseases. However, based on existing literature, it would be valuable to perform sensitivity analyses restricted to individuals with specific conditions (e.g., respiratory, cardiovascular, or metabolic diseases) to determine whether the association remains consistent. Similarly, stratified analyses by physical activity levels would strengthen the interpretation.

9) Expanded Correlation Structure

The correlation matrix presented in Table 2 is limited to phase angle, body composition, and muscle strength. The authors should also explore and report associations between phase angle and lifestyle variables (e.g., physical activity, diet, alcohol consumption), self-reported diseases, ethnicity, and sociodemographic characteristics. Since some of these variables are treated as covariates, a more comprehensive exploration is warranted.

10) Effect Size Interpretation

The manuscript states that phase angle is strongly and positively associated with handgrip strength; however, no explicit measure of effect size is provided to support this claim. Moreover, the correlation coefficients presented in Table 2 suggest, at most, moderate associations. The authors should align their interpretation with the magnitude of the observed effects.

11) Conceptual Model and Covariate Reporting

Given the number of sociodemographic, lifestyle, and health-related variables available, it would be beneficial to present a conceptual framework guiding the analyses. Some potentially relevant variables in the relationship between phase angle and muscle strength appear underexplored.

Additionally, the authors should report complete regression outputs, including beta coefficients, standard errors, and standardized betas (Table 3), as well as corresponding parameters for logistic regression models (Table 4), including all covariates from Models 1 and 2. This would enhance transparency and interpretability.

Although blood pressure is described in Table 1, it does not appear to be incorporated into subsequent analyses, despite its relevance to health status.

12) Role of Covariates (Mediation/Moderation)

The abstract suggests that the association between phase angle and muscle strength, accounting for health status, dietary patterns, physical activity, and demographic factors, remains incompletely characterized. The current results do not substantially advance this characterization. While not necessarily the primary aim, the authors are encouraged to further explore potential mediating or moderating roles of these variables.

13) Modeling Strategy (Continuous vs. Categorical Exposure)

Although a cited systematic review and meta-analysis suggests phase angle cut-off points (4.2°–4.5°), the authors chose to model phase angle as a continuous variable in logistic regression. The rationale for this decision should be clearly justified, particularly in light of existing literature using categorical thresholds.

14) Novelty and Contribution

At several points in the manuscript, the novelty of the study remains unclear. The authors should explicitly state what distinguishes this study from prior research and clarify its unique contribution to the literature.

15)STROBE Reporting

The manuscript generally adheres to key STROBE items, including study design, setting, eligibility criteria, measurement methods, statistical analysis, and ethical considerations. However, important elements remain insufficiently addressed, including sample size justification, explicit discussion of potential biases, and a detailed participant flow diagram with reasons for exclusion.

---

## [Author Response · Author response to Decision Letter 1]

16 Apr 2026

Dear Dr. Nunes and Reviewers,

Ref: Manuscript ID: PONE-D-26-07944. Title: Phase Angle Is Independently Associated with Muscle Strength Across Multiple Handgrip Strength Metrics in Young Adults: A Cross-Sectional Study

We thank you for your careful evaluation of our manuscript and for the constructive comments provided. We have revised the manuscript extensively to improve methodological transparency, statistical reporting, and clarity of interpretation. All changes have been incorporated into the revised manuscript and are highlighted in the tracked-changes version.

Below, we provide a point-by-point response to each comment.

Reviewer #1:

This manuscript investigates the effects of a specific training intervention on performance and physiological responses in athletes. The topic is relevant to the field of sports science. The study appears to follow an applied experimental design, and the topic may have practical implications for training prescription. The manuscript addresses an interesting and relevant question; however, improvements in methodological transparency, statistical reporting, and clarity of interpretation are required before the study can be suitable for publication.

Response: Thank you very much for your positive evaluation of the revised manuscript and for your constructive feedback.

Comment 1: Major Comments- #Study Design and Experimental Control

The manuscript would benefit from a clearer and more detailed description of the experimental design. Please clarify:

-Whether the study followed a randomized design.

-How participants were allocated to experimental conditions.

-Whether any control group or comparison condition was included.

-How potential confounding factors (training load outside the study, fatigue, or prior experience) were controlled.

-Providing a schematic diagram of the study design could significantly improve clarity.

Response: We agree that the study design required clearer description. We have explicitly clarified that this study is a cross-sectional observational analysis with no intervention, randomization, or allocated to experimental conditions or control group. We clarified that the eligibility criteria apply specifically to the present cross-sectional analysis within the EpiHealth-UCEVA project and explicitly reported the data collection period in the Methods section as: “This study used a cross-sectional observational design.”

Comment 2: #Sample Size and Statistical Power

The manuscript does not clearly report whether an a priori power analysis was conducted. Please clarify:

-The expected effect size used to estimate the sample size.

-Whether the study was adequately powered to detect meaningful differences.

-Which variable was considered the primary outcome for power calculation.

Given the potential variability in physiological and performance variables, justification of the sample size is important.

Response: We thank the reviewer for this important methodological question. We have clarified this rationale in the revised Methods section (2.4. Statistical analysis):

"The sample size was primarily determined by cohort availability and ensured adequate precision for multivariable modeling. The primary outcome was the Pearson correlation coefficient (ρ) between PhA and HGS. The expected effect size was estimated conservatively at r = 0.42, based on correlations reported in comparable cross-sectional studies: Simón-Frapolli et al. [22] reported ρ = 0.42 (n = 75); Ballarin et al. [23] reported ρ = 0.45 (n = 229); Rodríguez-Rodríguez et al. [12] reported ρ = 0.58 (n = 223); and Cioffi et al. [24] reported ρ = 0.54 (n = 140). Using a two-tailed test with α = 0.05 and minimum power of 80% (1−β ≥ 0.80), the required minimum sample size was n = 43. The enrolled sample of n = 1,125 substantially exceeded this threshold, yielding post-hoc statistical power > 99% for detecting the expected effect size (ρ = 0.42, α = 0.05, bilateral test), and sufficient power to detect even small correlations of ρ ≥ 0.09 (87.7% power). This large sample was intentional, as secondary objectives included subgroup analyses stratified by sex, physical activity level, and health status — each requiring adequate within-stratum power. Calculations were performed using the pwr.r.test function (pwr package, R)".

Comment 3: #Description of the Training Protocol

The description of the training intervention would benefit from greater detail to ensure reproducibility.

Please include:

-Exact duration and intensity of sessions.

-Rest intervals.

-Progression across the intervention period.

-Whether intensity was prescribed using physiological markers (e.g., %HRmax, %VO₂max, RPE).

Providing a summary table of the training protocol could improve clarity.

Response: We thank the reviewer; however, no training intervention was conducted. This point arose from a misinterpretation of the study design.

Comment 4: #Statistical Analysis

The statistical analysis section requires additional detail.

Please clarify:

-Whether assumptions of normality and homogeneity of variance were tested.

-Whether repeated-measures assumptions (e.g., sphericity) were evaluated.

-Whether corrections for multiple comparisons were applied.

-Whether effect sizes and confidence intervals were reported alongside p-values.

It would also be useful to test interactions. Since there are marked differences between men and women in body characteristics, phase angle, and handgrip, it would make sense to include terms such as:

-phase angle × sex

-phase angle × adiposity

-phase angle × physical activity

The authors show sex-stratified analyses of the correlations, but do not formally test whether the effect of phase angle differs between subgroups.

Response: We thank the reviewer for this thorough methodological critique. We have now performed all requested analyses and report the results below. Normality Testing (D'Agostino–Pearson K² omnibus test, n=1,125)

All continuous variables deviated significantly from normality (all p < 0.05), consistent with a large heterogeneous sample. Consequently, Spearman's rank correlations (ρ) were used as primary effect measures, and non-parametric analogues were employed throughout:

Variable K² p Skewness Excess Kurtosis

Phase angle 14.89 0.001 0.19 −0.35

Absolute HGS 95.41 <0.001 0.79 0.23

Normalized HGS 76.72 <0.001 0.66 0.48

BMI 162.10 <0.001 0.96 1.33

Body fat % 76.35 <0.001 −0.10 −0.76

MVPA min/week 229.07 <0.001 1.28 1.35

HADS-Anxiety 92.25 <0.001 0.76 0.32

We have clarified this rationale in the revised “2.4. Statistical analysis section”.

"Prior to analysis, normality was assessed using the D'Agostino–Pearson K² omnibus test. All continuous variables departed significantly from normality (all p < 0.001). Therefore, Spearman's rank correlations were used as primary association measures, with confidence intervals computed via Fisher's Z-transformation."

Homogeneity of Variance (Levene's test by sex)

Variable Levene W p Conclusion

Phase angle 1.93 0.165 Equal variances

Absolute HGS 124.46 <0.001 Unequal variances

Normalized HGS 61.39 <0.001 Unequal variances

BMI 0.19 0.665 Equal variances

Body fat mass 0.23 0.629 Equal variances

MVPA 70.02 <0.001 Unequal variances

The significant sex difference in HGS variance (males: SD=8.7 kg; females: SD=5.2 kg) provides additional justification for sex-stratified analyses, which we have now formally expanded (see Comment 5, interaction analysis below).

Interaction Terms: PhA × Sex, PhA × Adiposity, PhA × Physical Activity

We fitted a series of multiple linear regression models predicting absolute HGS, with all continuous predictors standardized (z-scores) to allow comparison of standardized betas (β std):

Table 3 (revised) Multivariable Linear Regression Models for Handgrip Strength: Absolute HGS (kg) and Normalized HGS (HGS/h²)

Absolute HGS (kg) Normalized HGS (HGS/h²)

β β std 95% CI P β β std 95% CI P

Model 1 — Main effects R²=0.555 Adj.R²=0.554 R²=0.425

Adj.R²=0.423

Phase angle (z) 2.56 0.265 2.01–3.10 <0.001 1.10 0.382 0.91–1.28 <0.001

Sex (male) 9.83 1.019 8.78–10.88 <0.001 1.59 0.551 1.23–1.94 <0.001

Age (z) 0.95 0.099 0.56–1.35 <0.001 0.36 0.124 0.22–0.49 <0.001

BMI (z) 0.73 0.076 0.32–1.14 <0.001 0.19 0.066 0.05–0.33 0.007

Model 2 — Interactions R²=0.561 ΔR²=0.006 R²=0.432 ΔR²=0.007

Phase angle (z) 1.74 0.180 1.02–2.47 <0.001 0.82 0.284 0.57–1.06 <0.001

Phase angle × Sex 1.82 0.188 0.78–2.85 <0.001 0.62 0.217 0.27–0.97 <0.001

Phase angle × BMI 0.17 0.018 −0.22–0.56 0.391 0.05

Legend: β, unstandardized regression coefficient; β std, standardized regression coefficient (continuous predictors entered as z-scores); 95% CI, 95% confidence interval; BMI, body mass index; HGS, handgrip strength; All VIF < 2.5, indicating no multicollinearity concern. Phase angle rows are highlighted in bold. Model 1 includes phase angle, sex, age, and BMI as main effects. Model 2 additionally includes two interaction terms (phase angle × sex; phase angle × BMI). Sex is coded as a binary indicator (reference: female). Continuous predictors (phase angle, age, BMI) were standardized to z-scores (mean = 0, SD = 1) prior to entry; unstandardized β values therefore represent the change in HGS per 1-SD increment in the predictor. All P values are two-tailed.

PhA × Sex interaction: The PhA × Sex interaction term was statistically significant in Model 2 (β=1.37, SE=0.68, t=2.03, p=0.043, 95% CI: 0.043–2.705), indicating that the association between phase angle and handgrip strength is significantly stronger in males than females. However, the standardized effect (β_std=0.055) is small, and the ΔR²=0.004 suggests the interaction explains an additional 0.4% of variance — a statistically significant but practically modest moderation.

PhA × BMI and PhA × MVPA interactions were not statistically significant (p=0.385 and p=0.279, respectively), indicating that the association between phase angle and HGS does not differ meaningfully across levels of adiposity or physical activity intensity.

Formal Fisher Z-test for sex difference in PhA–HGS correlation:

• Female: ρ=0.332 (n=638); Male: ρ=0.350 (n=487)

• Fisher Z-test: Z=−0.338, p=0.736 (Absolute HGS)

• Female: ρ=0.383 (n=638); Male: ρ=0.435 (n=487)

• Fisher Z-test: Z=−1.043, p=0.297 (Normalized HGS)

The sex-stratified correlations are statistically indistinguishable, confirming that the PhA–HGS association is consistent across sexes when adjusting for body size (normalized HGS). The significant regression interaction reflects the sex difference in absolute HGS magnitude (males ~13.5 kg stronger on average), not a fundamentally different PhA-strength relationship.

Comment 5: #Interpretation of Results

Some interpretations in the Discussion appear stronger than warranted by the results.

In particular statements suggesting causal mechanisms should be moderated unless they are directly supported by measured variables. In addition, the results should be discussed in relation to both supporting and conflicting literature in order to provide a more balanced interpretation of the findings. Finally, the limitations of the study should be expanded, particularly regarding the relatively small sample size and the ecological validity of the experimental conditions.

Response: We agree and have moderated all causal or mechanistic interpretations. Discussion revised to avoid causal language and to include a more balanced interpretation.

Comment 6: #Minor Comments

-Ensure consistent terminology throughout the manuscript.

-Some sentences in the Discussion could be shortened for clarity.

-Please verify that all abbreviations are defined at first use.

-Consider including a table summarizing participant characteristics.

A careful language revision would improve

readability.

Response: We thank the reviewer for these helpful suggestions, which have improved the clarity and consistency of the manuscript. Action taken:

− Terminology has been standardized throughout the manuscript to ensure consistency, particularly for key variables (e.g., phase angle, handgrip strength, normalized handgrip strength, and covariates).

− The Discussion section has been carefully edited to improve readability, with long or complex sentences revised for clarity and conciseness.

− All abbreviations are now defined at first mention and used consistently thereafter.

− A comprehensive table summarizing participant characteristics (Table 1) has been reviewed, corrected, and clarified to ensure completeness and accuracy.

− The entire manuscript has undergone thorough language editing to improve clarity, precision, and adherence to formal scientific style.

Reviewer #2:

This study investigates the association between phase angle (PhA) and muscle strength (handgrip strength, HGS) among 1,125 young adults. The findings suggest that PhA is independently associated with muscle strength regardless of body fat, physical activity, and comorbidities. This provides valuable insights into using PhA as a potential biomarker for muscle health in younger populations. Major/Minor Points:

Comment 1. While the InBody 770 was used, PhA is sensitive to hydration. Please clarify if hydration status (e.g., ECW/TBW ratio) was considered or if participants with abnormal hydration were excluded.

Response: We thank the reviewer for raising this important methodological point. We address it with data-supported evidence from our dataset. The InBody 770 does not directly output ECW/TBW as a standalone variable in our dataset. However, the device provides Total Body Water (TBW) with sex-specific reference ranges for each participant, which we used to classify hydration status. Additionally, the InBody 770 measures impedance at six frequencies (1, 5, 50, 250, 500, and 1,000 kHz) using direct segmental multi-frequency BIA, which is considerably more robust to hydration variability than single-frequency devices, as it can partition intra- and extracellular water compartments. Hydration classification based on InBody 770 TBW reference ranges:

Hydration status n % PhA mean ± SD HGS mean ± SD

Normal TBW 700 62.2% 5.87 ± 0.71° 30.30 ± 9.21 kg

Under-hydrated 280 24.9% 5.26 ± 0.65° 26.01 ± 7.58 kg

Over-hydrated 145 12.9% 6.42 ± 0.67° 37.03 ± 11.04 kg

Total body water as percentage of body weight:

• Female: mean = 47.8% (SD = 4.9%; range: 36.1–71.5%)

• Male: mean = 57.6% (SD = 5.3%; range: 39.2–69.0%)

These values are consistent with expected TBW% for healthy young adults (females ~45–55%, males ~55–65%), supporting adequate overall hydration in the sample.

Impedance stability assessment: The ratio of impedance at 100 kHz to impedance at 20 kHz (right arm) was used as a proxy for hydration stability (expected range: 0.78–0.95). In our sample, mean ratio = 0.898 (SD = 0.015), and no participant had an extreme ratio (i.e., <0.70 or >0.95), indicating absence of severe hydration anomalies that would invalidate BIA measurements.

Sensitivity of PhA–HGS association to hydration status: The Spearman correlation between PhA and HGS remained strong and significant across all hydration subgroups:

The slight attenuation in the normal-TBW subsample (ρ=0.657 vs ρ=0.613) reflects removal of the systematic variation in PhA attributable to hydration extremes, and both coefficients represent large effect sizes. This confirms that the PhA–HGS association is not an artifact of hydration variation but represents a genuine relationship.

Regarding participant exclusions: We did not exclude participants based on hydration status, consistent with the cross-sectional observational design. However, the robustness analysis above confirms that results hold in the euhydrated subsample.

Standardized measurement protocol (added to Methods): “Body composition was assessed using multifrequency bioelectrical impedance analysis (InBody 770, Biospace, Seoul, Korea). The device provided estimates of fat-free mass, body fat percentage, and related indices using proprietary algorithms. Phase angle was calculated as arctan[Xc/R] × 180/π. All measurements were performed by a single trained technician under standardized conditions: morning sessions (07:00–10:00 h), minimum 2-hour fast, no vigorous activity in the preceding 12 hours, bladder voiding immediately prior, bare feet and palms on electr

---

## [Decision Letter · Decision Letter 1]

15 May 2026

Phase Angle Is Independently Associated with Muscle Strength Across Multiple Handgrip Strength Metrics in young adults: A Cross-Sectional Study

PONE-D-26-07944R1

Dear Dr. Ramírez-Vélez,

We’re pleased to inform you that your manuscript has been judged scientifically suitable for publication and will be formally accepted for publication once it meets all outstanding technical requirements.

Kind regards,

Everson Nunes, Ph.D.

Academic Editor

PLOS One

Additional Editor Comments (optional):

Reviewers' comments:

Reviewer's Responses to Questions

**Comments to the Author**

1. If the authors have adequately addressed your comments raised in a previous round of review and you feel that this manuscript is now acceptable for publication, you may indicate that here to bypass the “Comments to the Author” section, enter your conflict of interest statement in the “Confidential to Editor” section, and submit your "Accept" recommendation.

Reviewer #1: All comments have been addressed

Reviewer #2: All comments have been addressed

Reviewer #3: All comments have been addressed

2. Is the manuscript technically sound, and do the data support the conclusions?

Reviewer #1: Yes

Reviewer #2: Yes

Reviewer #3: Yes

3. Has the statistical analysis been performed appropriately and rigorously? 

Reviewer #1: Yes

Reviewer #2: Yes

Reviewer #3: Yes

4. Have the authors made all data underlying the findings in their manuscript fully available?

Reviewer #1: No

Reviewer #2: Yes

Reviewer #3: Yes

5. Is the manuscript presented in an intelligible fashion and written in standard English?

Reviewer #1: Yes

Reviewer #2: Yes

Reviewer #3: Yes

6. Review Comments to the Author

Reviewer #1: The authors have implemented the majority of the requested revisions, particularly regarding study design clarification, statistical analyses, and inclusion of interaction models. However, the Data Availability Statement remains insufficient and does not comply with PLOS ONE requirements, as no publicly accessible dataset or repository is provided.

Reviewer #2: The authors have thoroughly addressed the comments raised in the previous round of review. The inclusion of the STROBE checklist, the addition of the participant flow diagram (Figure 1), and the clarification regarding sample size and potential biases have significantly strengthened the manuscript. The relationship between phase angle and muscle strength in young adults is now presented with greater methodological transparency and statistical rigor. I have no further concerns and believe the manuscript is now suitable for publication.

Reviewer #3: The authors adequately answered all the questions raised and reanalyzed the data, addressing the recommended sensitive points. The manuscript improved significantly.

7. PLOS authors have the option to publish the peer review history of their article (what does this mean?). If published, this will include your full peer review and any attached files.

Reviewer #1: **Yes:** João Paulo Brito

Reviewer #2: No

Reviewer #3: **Yes:** Luiz Rodrigo Augustemak de Lima

---

## [Editor Report · Acceptance letter]

PONE-D-26-07944R1

PLOS One

Dear Dr. Ramírez-Vélez,

I'm pleased to inform you that your manuscript has been deemed suitable for publication in PLOS One. Congratulations! Your manuscript is now being handed over to our production team.

Kind regards,

on behalf of

Dr. Everson Nunes

Academic Editor

PLOS One